# Investigating the uptake, effectiveness and safety of COVID-19 vaccines: protocol for an observational study using linked UK national data

Eleftheria Vasileiou [1], Ting Shi [1], Steven Kerr,[1] Chris Robertson,[2,3] Mark Joy,[4] Ruby Tsang [4], Dylan McGagh,[4] John Williams,[4] Richard Hobbs,[4] Simon de Lusignan [4], Declan Bradley,[5] Dermot OReilly,[5] Siobhan Murphy,[5] Antony Chuter,[6] Jillian Beggs,[6] David Ford,[7] Chris Orton [7], Ashley Akbari [7], Stuart Bedston,[7] Gareth Davies,[7] Lucy J Griffiths [7], Rowena Griffiths,[7] Emily Lowthian [7], Jane Lyons [7], Ronan A Lyons [7], Laura North,[7] Malorie Perry,[8] Fatemeh Torabi,[7] James Pickett,[9] Jim McMenamin,[3] Colin McCowan,[10] Utkarsh Agrawal,[10] Rachael Wood [1,11], Sarah Jane Stock [1,11], Emily Moore,[3] Paul Henery [12], Colin R Simpson,[1,13] Aziz Sheikh [1]

For numbered affiliations see end of article.

**Correspondence to**
Dr Eleftheria Vasileiou;
eleftheria.vasileiou@ed.ac.uk

## ABSTRACT

**Introduction** The novel coronavirus SARS-CoV-2, which emerged in December 2019, has caused millions of deaths and severe illness worldwide. Numerous vaccines are currently under development of which a few have now been authorised for population-level administration by several countries. As of 20 September 2021, over 48 million people have received their first vaccine dose and over 44 million people have received their second vaccine dose across the UK. We aim to assess the uptake rates, effectiveness, and safety of all currently approved COVID-19 vaccines in the UK.

**Methods and analysis** We will use prospective cohort study designs to assess vaccine uptake, effectiveness and safety against clinical outcomes and deaths. Test-negative case–control study design will be used to assess vaccine effectiveness (VE) against laboratory confirmed SARS-CoV-2 infection. Self-controlled case series and retrospective cohort study designs will be carried out to assess vaccine safety against mild-to-moderate and severe adverse events, respectively. Individual-level pseudonymised data from primary care, secondary care, laboratory test and death records will be linked and analysed in secure research environments in each UK nation. Univariate and multivariate logistic regression models will be carried out to estimate vaccine uptake levels in relation to various population characteristics. VE estimates against laboratory confirmed SARS-CoV-2 infection will be generated using a generalised additive logistic model. Time-dependent Cox models will be used to estimate the VE against clinical outcomes and deaths. The safety of the vaccines will be assessed using logistic regression models with an offset for the length of the risk period. Where possible, data will be meta-analysed across the UK nations.

## Strengths and limitations of this study

► We will use national data for each UK nation and across the UK general population.
► Rapid and robust real-time estimates on uptake, effectiveness and safety of COVID-19 vaccines will be provided using data from existing national pandemic platforms in the UK.
► This is an observational study and analyses are, therefore, potentially susceptible to residual or unmeasured confounders.

**Ethics and dissemination** We obtained approvals from the National Research Ethics Service Committee, Southeast Scotland 02 (12/SS/0201), the Secure Anonymised Information Linkage independent Information Governance Review Panel project number 0911. Concerning English data, University of Oxford is compliant with the General Data Protection Regulation and the National Health Service (NHS) Digital Data Security and Protection Policy. This is an approved study (Integrated Research Application ID 301740, Health Research Authority (HRA) Research Ethics Committee 21/HRA/2786). The Oxford-Royal College of General Practitioners Clinical Informatics Digital Hub meets NHS Digital's Data Security and Protection Toolkit requirements. In Northern Ireland, the project was approved by the Honest Broker Governance Board, project number 0064. Findings will be made available to national policy-makers, presented at conferences and published in peer-reviewed journals.

## INTRODUCTION

The first human cases infected by the novel coronavirus SARS-CoV-2 pathogen were

detected in Wuhan, China in December 2019.[1] On 11 March 2020, WHO declared the COVID-19 as a global pandemic, which as of 20 September 2021 has caused more than 228 million infections and four million deaths worldwide.[1] The need for vaccines against this novel virus triggered an emergency response by governments, pharmaceutical companies and research institutions to develop, licence and manufacture COVID-19 vaccines at scale. Dozens of COVID-19 vaccines are currently under development with some vaccines now having successfully completed their prelicensure clinical trials and been approved for population vaccine administration.[2 3] The speed with which the world's first vaccines[3] were available for mass administration at the end of 2020 is unprecedented given that typically it takes years for a vaccine to be available for use at a population level.[4]

The Pfizer-BioNTech and Oxford-AstraZeneca were among the first vaccines approved by national regulatory authorities such as the UK's Medicines and Healthcare products Regulatory Agency.[5] Based on prelicensure clinical trials, the Pfizer-BioNTech vaccine was 95% effective at preventing laboratory-confirmed COVID-19 illness in individuals without evidence of previous infection.[6] The Oxford-AstraZeneca vaccine has also reported significant efficacy of 64% and 70% after one and two doses, respectively, against symptomatic disease.[7] As of the 8 January 2021, the UK has also approved a third COVID-19 vaccine manufactured by Moderna which has shown 94% efficacy against confirmed SARS-CoV-2 infection after receipt of second dose.[8] All three vaccines were well tolerated with mild-to-moderate side effects mostly reported.[6–8] In March 2021, the European Medicines Agency reported extremely rare but serious side effects including blood clots and bleeding following administration of the Oxford-AstraZeneca vaccine.[9 10] As a result, the Joint Committee on Vaccination and Immunisation (JCVI) of the UK Government has recommended that healthy adults aged 18–39 years old should not be offered the Oxford-AstraZeneca vaccine but the Pfizer-BioNTech or Moderna vaccines instead.[11] Rare cases of cardiac inflammation including myocarditis and pericarditis following immunisation with the Pfizer-BioNTech and Moderna vaccines have also been reported[12] and are thus closely monitored by national and international regulatory agencies.

As of 20 September 2021, over 48 million people have received their first vaccine dose and over 44 million people have received their second vaccine dose across the UK[13] based on JCVI's vaccination priority list which targets those most at risk of severe COVID-19 illness (eg, older adults and people with comorbidities).[14]

Timely postlicensure monitoring of coverage, protection and safety of these newly introduced vaccines is imperative.[15] Specifically, robust observational epidemiological studies are required to measure coverage rates in the population in relation to demographic and other population characteristics, assess effectiveness against infection, severe illness and deaths, and to detect adverse events. These postmarketing observational studies will add additional value to the prelicensure clinical trials as they can assess real-life effects of the COVID-19 vaccines and the impact of the vaccination programme at population levels.[15] All UK-licensed vaccines have demonstrated high efficacy in clinical trials; however, more evidence is needed about the type, level and duration of protection for different segments of the population. The recommended time period between administration of the first and second doses of the Pfizer-BioNTech[6] and Oxford-AstraZeneca[7] vaccines are 3 and 6 weeks, respectively. The UK government decided to lengthen this gap to up to 12 weeks for both vaccines.[14] This is because of the desire to provide some degree of protection to as many people as possible and the decision was supported by findings from the Oxford-AstraZeneca vaccine trial, which have shown that vaccine efficacy was higher (65%) in the group that was administered the second dose more than 6 weeks after the first dose, compared with the group given the second dose less than 6 weeks after the first dose (53%).[7 16] A simulation study has also shown that high vaccine coverage even with less efficacious vaccines (due to partial immunisation) can lead to a greater reduction of SARS-CoV-2 infection levels compared with a lower vaccine coverage with more efficacious vaccines.[17] On the other hand, some experts have expressed their concerns that delay between doses could increase the risk of vaccine-resistant strains emerging due to a partially immunised population.[16] The provision of timely estimates on the protection conferred between doses is thus urgently needed. The assessment of vaccine-induced adverse events also needs to be carried out, particularly for rare adverse events that are usually only detectable in large population studies.[18] The rate of vaccine receipt by demographic, socioeconomic and other epidemiological characteristics also needs to be measured. In the UK, the 'Understanding Society' COVID-19 survey asked 12 035 participants (in November 2020) their likelihood of vaccine uptake and reason for hesitancy.[19] High levels of hesitancy were found in women (21%), younger adults aged 16–24 years old (27%), those with lower education levels (19%) and in black (72%) and Pakistani/Bangladeshi (42%) ethnic groups.[19] This is particularly concerning given that ethnic minority groups are some of the subgroups of the population that have been most at risk in this pandemic.[19] The identification of key factors related to vaccine uptake may be useful in efforts to increase uptake and subsequently maximise the impact of the COVID-19 vaccination programme in the UK.

### Aim and objectives

The aims of this study are to assess the uptake, effectiveness, and safety of the currently licensed COVID-19 vaccines (Pfizer-BioNTech, Oxford-AstraZeneca and Moderna) in the UK using linked healthcare and administrative data. We will also seek to assess any additional licensed vaccines during the course of this study.

Our primary objectives are to: (1) measure variation in vaccine uptake in relation to population characteristics; (2) assess vaccine effectiveness (VE) against infection, transmission,[20] severe outcomes and deaths; and (3) identify the risk of adverse events following immunisation (AEFIs) in each UK nation. Our secondary objectives are to provide UK-wide pooled estimates of each primary objective.

## METHODS
### Study design
A prospective cohort study design will be used to measure variations in vaccine uptake and assess VE against severe illness, deaths, secondary SARS-CoV-2 infection due to household transmission. A test-negative design (TND) case-control study will be carried out to assess VE against laboratory confirmed COVID-19 infection. In the TND, cases and controls will be those with a positive and negative test for COVID-19, respectively.[21] A self-controlled case series (SCCS) study design will be used to assess the risk of AEFI. The SCCS study will be used to determine the relative incidence of adverse events for exposed time periods (periods following vaccine administration) compared with unexposed time periods (prevaccination or postvaccination periods unrelated to vaccination) in individuals who present with the outcome of interest.[22] For more severe and event dependent safety outcomes, a retrospective cohort study will be considered.

### Overview of linked databases and study population
We will use pseudonymised individual level data routinely collected at primary and secondary care visits, linked with mortality, laboratory and vaccination data across the UK. Unique national datasets will be developed and hosted within secure research environments in each UK nation with standardised individual-level analyses run across datasets. Pooled estimates across the UK nations will also be calculated. For England, data from the Royal College of General Practitioners Research and Surveillance Centre (RCGP RSC) (approximately 5.4 million people)[23] will be used. We will also access and analyse national coverage data from Northern Ireland (approximately 1.9 million people). For Scotland, data will be derived from the Early Pandemic Evaluation and Enhanced Surveillance of COVID-19 (approximately 5.4 million people).[24] For Wales, data from the Controlling COVID-19 through enhanced population surveillance and intervention (ConCOV) (approximately 3.2 million people)[25 26] and the Secure Anonymised Information Linkage (SAIL) Databank[27] will be used. See table 1 for details on data sources from each UK nation.

### England
English primary care data will be held in the Oxford-Royal College of General Practitioners Clinical Informatics Digital Hub.[28] Data will be pseudonymised as close to source as possible using an National Health Service

(NHS) Digital Data Security and Protection approved method.[29 30] NHS Digital will link additional data to these pseudonyms.

### Northern Ireland
For the Northern Irish data, the Honest Broker Service (HBS) will carry out the data linkage and offer a secure environment for researchers to access and analyse the pseudonymised individual-level data. The Healthcare Number will be used to link individuals' data though replaced with an anonymous study ID in the analysis dataset.

### Scotland
For the Scottish data, Public Health Scotland (PHS) will carry out the data linkage and offer a secure environment for researchers to access and analyse the pseudonymised individual level data.[24] The Community Health Index number (a unique identifier for each resident receiving healthcare) will be used to link individuals' data which will be replaced by a study ID.

### Wales
Withing SAIL and in collaboration with the ConCOV project,[25 26] data will be linked and anonymised from NHS sources via a mature split-file system. All identifiable data will stay within the NHS, and will be linked to the Welsh Demographic Service Dataset (WDSD), pseudonymised and encrypted within NHS Wales by Digital Health and Care Wales, with pseudonymised demographics supplied to SAIL at Swansea University,[27] where the non-identifiable clinical data are held. SAIL will link the demographic data to relevant clinical data, and then further encrypt the linked data before presenting to research teams within a secure virtual desktop.

### Exposure (vaccination) data
#### England
In England, data on vaccination will derive from general practitioners (GPs) and the National Immunisation Management Service (NIMS).[31] NIMS is the System of Record for the NHS COVID-19 vaccination programme in England developed by NHS Digital.[31] NIMS will collect any demographic, GP and employee (for NHS) data to identify groups of the population eligible for vaccination. Data collected from NIMS will also feed back into GP systems so that an individual's electronic health record is updated regarding to their vaccination history.[31] Vaccination data in GP records will be recorded using the Systematised Nomenclature of Medicine (SNOMED) Clinical Terms (CT).[32] RCGP RSC now uses SNOMED CT for all its key variables including vaccine data.[33] A key part of these curated variables are those for COVID-19; there have been three iterations of these and we have carefully curated case definitions.[34–36]

#### Northern Ireland
In Northern Ireland, GP practices and Trust vaccination sites are currently delivering the COVID-19 vaccination

**Table 1** Available UK datasets for each data item of interest

| Data item | England | Northern Ireland | Scotland | Wales |
|---|---|---|---|---|
| **Exposures** | | | | |
| Pfizer-BioNTech vaccine | GP, NIMS | VMS | GP, TVMT/PHS, SIRS | CVVD |
| Oxford-AstraZeneca vaccine | GP, NIMS | VMS | GP, TVMT/PHS, SIRS | CVVD |
| Moderna vaccine | GP, NIMS | VMS | GP, TVMT/PHS, SIRS | CVVD |
| **Outcomes** | | | | |
| Laboratory confirmed SARS-CoV-2 infection | GP, Pillar 1 and 2 SGSS, PHE | Pillar 1 and Pillar 2 | ECOSS | PATD (Pillar 1, 2 and 3 data from all NHS and private labs), CVLF testing and results data |
| COVID-19-related GP consultation | GP | NA | GP | WLGP |
| COVID-19-related emergency department consultation | GP, ECDS | Symphony, NIRAES | SMR01 | EDDD and EDDS |
| COVID-19-related hospital admission | GP, SUS | Admissions and discharge dataset | SMR01 | PEDW |
| COVID-19-related ICU admission | GP, CHESS | Admissions and discharge dataset | SICSAG | CDDS, ICCD and ICNC |
| COVID-19-related death | GP, ONS, SSRS | NHAIS | NRS | ADDE and ADDE (ONS mortality), CDDS and WDSD |
| Secondary SARS-CoV-2 infection due to household transmission | RCGP RSC household key | Pillar 1 and Pillar 2 dataset | ECOSS | PATD (Pillar 1, 2 and 3 data from all NHS and private labs), CVLF testing and results data |
| Maternity outcomes | GP, MSDS | NIMATS | COPS study | ADBE (ONS births), MIDS and NCCH |
| **Patient characteristics and confounders** | | | | |
| Age | GP | NHAIS | GP | C19_COHORT20 |
| Sex | GP | NHAIS | GP | C19_COHORT20 |
| Socioeconomic status | Postal code to IMD | NHAIS | GP | C19_COHORT20 |
| Ethnicity | GP, SUS | VMS | Census 2011 | National ethnicity spine (made up of 20 EHR data sources and the ONS Census 2011) |
| Underlying medical condition | GP | EPD | GP | GP, WLGP, PEDW, CVSP |
| Type of settlement (urban/rural) | GP | NHAIS | GP | C19_COHORT20 |
| Type of settlement (eg, private home, care home or social housing) | GP | NHAIS | GP | C19_COHORT20, CARE |
| Smoking status | GP | NA | GP | WLGP |
| Body Mass Index | GP | NA | GP | WLGP |
| Prescribed medications | GP | EPD | GP, PIS, HEPMA | WLGP, WDDS |
| Other non-COVID-19 vaccines (eg, influenza, pneumococcal) | GP | VMS | GP | WLGP, NCCH |
| Occupation (eg, healthcare workers, front-line workers, essential workers) | GP where recorded | Pillar 1 and 2 | To be confirmed | HWRA, SWAC |
| History of healthcare utilisation (eg, GP consultations, hospital admissions) | GP, SUS | Admissions and discharge dataset | GP, SMR01 | PEDW, WLGP |

ADBE, Annual District Birth Extract (ONS Births); ADDE, Annual District Death Extract (ONS Deaths); CDDS, Critical Care DataSet; CHESS, COVID-19 Hospitalisation in England Surveillance System; COPS, COVID-19 in Pregnancy in Scotland; CVSP, COVID-19 Shielded People list; CVVD, COVID-19 Vaccine Data; ECDS, Emergency Care Data Set; ECOSS, Electronic Communication of Surveillance in Scotland; EDDD, Emergency Department Data Daily; EDDS, Emergency Department Dataset; EPD, Electronic Prescribing Database; GP, general practitioner; HEPMA, Hospital Electronic Prescribing and Medicines Administration; HES, Hospital Episode Statistics; ICU, intensive care unit; IMD, Index of Multiple Deprivation; LIS, Laboratory Information System; MIDS, Maternal Indicators DataSet; MSDS, Maternity Services Dataset; NCCH, National Community Child Health database; NHAIS, National Health Applications and Infrastructure Services; NHS, National Health Service; NIMATS, Northern Ireland Maternity System; NIMS, National Immunisation Management Service; NIRAES, Northern Ireland Regional Accident and Emergency System; NRS, National Records of Scotland; ONS, Office for National Statistics; PATD, Pathology data COVID-19 Daily; PEDW, Patient Episode Database for Wales; PHE, Public Health England; PHS, Public Health Scotland; PIS, Prescribing Information System; RCGP RSC, Royal College of General Practitioners Research and Surveillance Centre; SGSS, Second Generation Surveillance System; SICSAG, Scottish Intensive Care Society Audit Group; SIRS, Scottish Immunisation and Recall System; SMR01, Scottish Morbidity Record 01; SUS, Secondary Users Service; TVMT, Turas Vaccination Management Tool; VMS, Vaccine Management System; WDSD, Welsh Demographic Service Dataset; WLGP, Welsh Longitudinal General Practice.

programme. Vaccination data from GP practices and Trust vaccination sites are stored in a central Vaccination Management System and made available via the HBS secure research environment.[37]

**Scotland**

In Scotland, GPs usually facilitate vaccination programmes and record any data related to vaccine administration. GPs use Read codes to code and record relevant information

arising from a patient consultation.[24] Additional data on vaccination will also be available via the Turas Vaccination Management Tool (TVMT), which is a web-based tool that enables front-line vaccinators to capture and create real-time patient vaccination records. PHS is currently collating vaccination data from Turas.[38] Scheduled vaccinations for children may also be recorded in the Scottish Immunisation Recall System (SIRS) database. Data on vaccinations administered at schools and not in GPs may thus derive from SIRS.[24]

### Wales

In Wales, the vaccine programme is administered and recorded nationally in the all Wales Immunisation System (WIS) and is available in SAIL through the COVID-19 Vaccine Data. This is a separate independent system and data source to the GP data, which is also available in SAIL with vaccination recorded using Read codes.[27]

## Outcome data
### Laboratory confirmed outcomes
#### England

For England, most community testing (called Pillar 2) is in the GP record, though for this study we will additionally link to resources held by NHS Digital this is the Second Generation Surveillance System, this also included hospital tests.[39]

#### Northern Ireland

For Northern Ireland, the Pillar 1 dataset is extracted from the Laboratory Information Systems in Northern Ireland hospitals on a daily basis into a central repository in the Health and Social Care (HSC) Regional Data Warehouse, which is maintained by the HSC Business Services Organisation.[39] It contains details of COVID-19 antigen tests carried out in each of the hospital laboratories, including those processed on behalf of primary care, social care and community settings. Pillar 2 data are processed by NHS Digital and extracts for NI residents are sent to the NI HSC Regional Data Warehouse.[39]

#### Scotland

For Scotland, laboratory results from Scottish diagnostic and reference laboratories are captured by the Electronic Communication of Surveillance in Scotland (ECOSS) database, which can be used for surveillance and research purposes.[24] Laboratory results from NHS and community (Lighthouse laboratory) testing centres will also be available through ECOSS.

#### Wales

For Wales, national coverage of Pillar 1, 2 and 3 data from all NHS and private laboratories will be available, as well as national lateral flow testing data.[39]

We will also pursue to access genome sequencing data in a proportion of laboratory tests positive for SARS-CoV-2 available from national sequencing centres where possible.

### Clinical outcomes

Data on primary care consultations for COVID-19 illness will be accessed via GP electronic health records in each UK nation.

#### England

In England, we will link to secondary care data through collections held by NHS Digital. Hospital data are held in two forms: Hospital Episode Statistics (HES), which are the long-term validated record; and Secondary Uses Services which is an extract of contemporary operational data which after validation will become HES. We plan to use HES and also have an extract of intensive care data. We will access these data through NHS Digital's Data Access Request Service.[40] We also have SARS-CoV-2 virology data that were collected through the sentinel surveillance system.[41] The English primary care sentinel system within RCGP RSC has a strong working relationship with Public Health England (PHE), with whom we have worked closely for over half a century.[42]

#### Northern Ireland

In Northern Ireland, the Admissions and Discharge dataset will be used to collate data relating to admitted patient care delivered by HSC hospitals in Northern Ireland, generated by the patient administration systems within each hospital. These data are held centrally by the HSC Regional Data Warehouse.[43]

#### Scotland

In Scotland, data on patients receiving care in general or acute hospitals are recorded in the Scottish Morbidity Record 01 (SMR01).[24] The International Statistical Classification of Diseases and Related Health Problems, 10th Revision (ICD-10) codes are used to index any diagnoses recorded in the patient's medical notes by a clinician.[24] Consistent and high level (>90%) of data accuracy have been shown in recent data quality reports for the SMR01 database.[20] Data on adult patients admitted to general intensive care units (ICUs) will derive from the Scottish Intensive Care Society Audit Group database.[24]

#### Wales

In Wales, data on all interactions with secondary care including Accident and Emergency (A&E) events (Emergency Department Dataset), inpatient hospital admissions (Patient Episode Database for Wales), intensive care (Critical Care DataSet—CDDS and Intensive Care National Audit and Research Centre) and outpatient appointments and activity (Outpatient Database for Wales) are all available within SAIL Databank.[27] There are also a collection of specialised services and condition specific secondary care such as cancer which are available within SAIL.[27]

### Deaths
#### England

In England, NHS Digital's Personal Demographic Service flags the date of death in the GP record. We have previously

used these data of death to report excess mortality, both overall[44 45] and in people with known COVID-19 status.[46] We will augment these data with the certificated cause of death provided by the Office for National Statistics (ONS), which will be linked for us at individual pseudonymised patient level by NHS Digital.

### Northern Ireland

In Northern Ireland, the National Health Applications and Infrastructure Services (NHAIS) will be used for mortality data.[47] NHAIS receives regular updates from General Register Office on fact and cause of death. ICD-10 codes on deaths records derive from diagnoses recorded by the certifying doctor on the death certificate.[47]

### Scotland

In Scotland, the death registry within National Records of Scotland records information included in the death certificates.[24] ICD-10 codes on deaths records derive from diagnoses recorded by the certifying doctor on the death certificate.[24]

### Wales

In Wales, multiple sources of mortality data including information from the National Population Spine (WDSD), ONS death data (ADDE and ADDD) and a national NHS master patient index record (Consolidated Death Data Source—CDDS) will be accessed to retrieve complete, harmonised details on cause and associated mortality details.[27]

### Pregnancy and neonatal outcomes

Capturing vaccine exposure in pregnancy is important.[48] As no trial to date has included pregnant women this type of study is the only opportunity to explore safety and effectiveness in pregnant women and their babies.

### England

In England, we will use a customised 'sliding window' to capture pregnancy data.[49] Specifically, an algorithm that accurately inferred pregnancies will be used by adopting an ontological approach for case finding.[49] The ontological approach will thus be used to identify pregnancies and associated complications using a systematic approach to derive this information from routinely collected administrative health data which will be available via the RCGP RSC.[49]

### Northern Ireland

In Northern Ireland, data will be accessed via the Northern Ireland Maternity System.[50]

### Scotland

In Scotland, pregnancy and neonatal outcomes in Scottish participants will be identified through the COVID-19 in Pregnancy in Scotland study.[51]

### Wales

In Wales, pregnancy and neonatal outcomes will be identified from linked Annual District Birth Extract, National Community Child Health and Maternity Indicators DataSet. Data sources for Wales will be sought via the SAIL Databank.[27]

### Exposure definitions

Data on the currently licensed COVID-19 vaccines, including Pfizer-BioNTech, Oxford-AstraZeneca and Moderna, will be derived from GPs, NIMS, TVMT/PHS, WIS and HSC Trusts databases. For the first vaccine dose (partial vaccination), an individual will be defined as exposed or vaccinated from day 14 after receiving the first dose between the period of 8 December 2020 and until the end of follow-up. For the second vaccine dose (full vaccination), an individual will be defined as exposed or vaccinated from day 14 after receiving the second dose during the study period. Exposed or vaccinated groups will be stratified by the following time intervals: (1) 0–13 days after dose 1; (2) 14–20 days after dose 1; (3) 21–27 days after dose 1; (4) 28–34 days after dose 1; (5) 35–41 days after dose 1; (6) >42 days after dose 1; (7) 0–13 days after dose 2 and (8) >14 days after dose 2.

Controls or unvaccinated will be defined as those who have not yet received a COVID-19 vaccine or have only received one vaccine dose. Controls who become vaccinated with any vaccine (ie, including one dose of Moderna) or receive a second vaccine will then be assigned within the exposure group. As a result, follow-up of the exposure period will be censored for both the vaccinated and control recipient if the control meets the criteria to be classified as exposed (receiving a first dose when compared with the unvaccinated group and receiving a second dose when compared with the partially/one dose vaccinated group). Maximum follow-up period will correspond to the latest event date depending on the outcome of interest. Similar vaccinated and unvaccinated groups and periods will be determined for the Moderna vaccine.[52]

### Outcome definitions
#### Adverse events following immunisation

Adverse events to be monitored are derived from the safety results of the prelicensure vaccine clinical trials, common side effects related to influenza vaccines and an unpublished study protocol of an ongoing observational study.[6 7 53 54] These include use of health services such as GP or out-of-hours GP consultation, A&E department attendance, inpatient hospital admission and admission to ICU for suspected adverse events. Safety of vaccines in pregnant women will also be considered once vaccines are widely administered in this group of the population. AEFI by specific vaccine type will also be considered. A full list of candidate AEFI is available in in online supplemental material, appendices 1 and 2.

### VE outcomes

Protective effects of the vaccines will be assessed against the following outcomes: (1) RT-PCR laboratory confirmed COVID-19 infection; (2) GP consultations related to

suspected or confirmed COVID-19 illness; (3) A&E attendance related to suspected or confirmed COVID-19 illness at presentation; (4) hospital admissions related to confirmed COVID-19 illness; (5) ICU admissions related to confirmed COVID-19 illness and (6) deaths related to suspected or confirmed COVID-19 illness. We will also explore the effects of vaccines on secondary SARS-CoV-2 infection due to household transmission. VE against maternal and neonatal COVID-19 related outcomes will be explored once the vaccines are more widely available to pregnant women.

A COVID-19 hospital or ICU admission will be defined based on either a RT-PCR confirmed positive test for SARS-CoV-2 in the 28 days prior to admission or based on an ICD-10 code for COVID-19 (U07.1 or U07.2) in any diagnostic position. A COVID-19 death will be defined as COVID-19 as the underling ICD-10 cause of death recorded on the death certificate, or death from any cause within 28 days of a positive RT-PCR test for SARS-CoV-2.

## Population characteristics, confounding factors and effect modifiers

A number of key characteristics that could explain variation in vaccine uptake will be considered. In addition, these characteristics could confound our planned analyses. We will determine and include these characteristics or potential confounders at the baseline of our study's cohort which include (see table 1 for details on data sources for each UK nation):

### Sociodemographics

Sex at birth will be included in a binary format (females and males). Age will be included in bands that will be determined based on available vaccination data. Socio-economic status will be assessed by the following national versions of area level deprivation indices: (1) Index of Multiple Deprivation (IMD), 2019 version for England[55 56]; (2) the Scottish IMD (SIMD)[24]; (3) the Welsh IMD[57] and (4) the Northern Ireland Multiple Deprivation Measure.[58] For England, we will maximise the capture of ethnicity data through the use of a customised ontology.[59] Ethnicity data for the other UK nations will also be included if available.

### Geographic

In England, the RSC will use the ONS data on population density and classify households as rural, urban (town and city) or conurbation.[60] The RSC also has a unique household key, so we can report household incidence of respiratory and other infectious conditions.[61] In Scotland, settlement type will be included using the urban/rural sixfold classification (UR6).[24] In Wales, urban and rural household classification that is based on the Lower-layer Super Output Area of the person's residence information will be used.[26] In Northern Ireland, NISRA's Statistical Classification and Delineation of Settlements will be used to determine settlement type.[62] Type of residence will also be considered such as private residence, care home and social/council housing for each UK nation if data are available.

### Smoking status

Smoking status will be included through four categories: smoker, ex-smoker, non-smoker and 'not recorded'. Smoking status will not be determined for Northern Ireland due to no access to primary care data.

### At-risk underlying medical conditions

Based on the QCOVID algorithm,[63] we will consider the following conditions: (1) cardiovascular conditions (atrial fibrillation, heart failure, stroke, peripheral vascular disease, coronary heart disease, congenital heart disease); (2) diabetes (type 1 and type 2); (3) respiratory conditions (asthma, rare respiratory conditions (cystic fibrosis, bronchiectasis or alveolitis), chronic obstructive pulmonary disease, pulmonary hypertension or pulmonary fibrosis); (4) cancer (blood cancer, chemotherapy, lung or oral cancer, marrow transplant, radiotherapy); (5) neurological conditions (cerebral palsy, Parkinson's disease, rare neurological conditions (motor neuron disease, multiple sclerosis, myasthenia, Huntington's chorea), epilepsy, dementia, learning disability, severe mental illness) and (6) other conditions (liver cirrhosis, osteoporotic fracture, rheumatoid arthritis or systemic lupus erythematosus, sickle cell disease, venous thromboembolism, solid organ transplant, renal failure (chronic kidney disease stages 3–5 with or without dialysis or transplant).[63] Body mass index will also be considered.[63]

### History of healthcare utilisation

Number of GP consultations and hospital admissions in the 6 months before the start of the study cohort (December 1, 2020) will also be measured as a proxy of severity of pre-existing medical conditions.

### History of non-COVID-19 vaccination

Receipt of influenza or pneumococcal vaccination during the 2020–2021 season (1 September to 30 November 2020) available in a binary format (yes or no) will be included which could be predictive of COVID-19 vaccination.

### History of prescribed medications

Based on the QCOVID algorithm,[63] we will measure number (≥4) of prescriptions from general practices for oral steroids, long acting β agonists or leukotrienes, immunosuppressants in previous 6 months prior to the start of the study cohort (1 December 2020). Prior or concomitant usage of the novel oral anticoagulants, warfarin and heparin will also be measured.

### History of COVID-19 infection

Any suspected or confirmed COVID-19 infections in the previous last 6 months (prior to 1 December 2020) will be included.

## Statistical analysis

We will report summary statistics of baseline characteristics for the individuals in the study as of 1 December

2020. These will be also stratified by vaccine status and respective study outcome of interest. Relevant measures of tendency (eg, mean, median, proportion) and variability (eg, SD, IQR) will be calculated. Relative estimates (eg, OR and rate ratio (RR)) and their respective 95% CI for risk of outcomes of interest will also be included for different population strata. Missing data will be reported as percentages of total, imputation will be considered if possible, and sensitivity analysis will be carried out to examine if the nature of the missing data mechanism affects the study findings. All statistical hypotheses tests will be two tailed with significance level set at 5% for all outcomes of interest. Statistical computing language R will be used to conduct all planned statistical analyses.

### Individual patient-level analyses
#### Vaccine uptake
*Prospective cohort study for vaccine uptake*

Overall proportion of individuals that receive the vaccine, stratified by sociodemographic, medical and other characteristics will be reported using a prospective cohort study. We will also consider reporting uptake levels within certain subgroups of the population where possible (eg, ethnic minorities and healthcare and other front-line workers, care home residents and pregnant women). We will also seek to examine patterns related to the number of eligible individuals that were offered vaccine but refused to be vaccinated if data are available. Univariable and multivariable logistic regression models will be carried out to estimate the coefficient of each predictor variable in the model and their 95% CIs, as well as the OR for vaccine uptake.

#### Vaccine effectiveness
*TND case–control study for laboratory-confirmed outcomes*

Vaccine status will be compared between cases (patients with a positive test for COVID-19) and controls (patients with a negative test for COVID-19) using a TND case–control study.[24] The main advantage of the TND studies compared with traditional case–control studies is that it minimises confounding factors from health care-seeking behaviour, which means both cases and controls have similar likelihood of seeking healthcare when having symptoms indicative of COVID-19 illness.[64] Selection bias can still arise if study participants are not recruited based on predefined criteria (eg, signs/symptoms indicative to COVID-19 illness) but based on clinician-ordered test.[64] In this scenario, clinicians may be more likely to carried out a test on patients that are more likely to have COVID-19 illness (outcome) or not being vaccinated.[64] This will result to biased sampling (non-representativeness) of the study participants from the source population which could lead to overestimation of the VE estimates.[64] VE is estimated based on the OR using the formula VE=1 OR. OR is defined as the odds of a SARS-CoV-2 infection among the vaccinated group divided by the odds of a SARS-CoV-2 infection among the unvaccinated group. A generalised additive logistic

model will be used to estimate the OR and VE. Stratified VE estimates by vaccine type, dose, dosing schedules, COVID-19 strain, age group and underlying medical condition will also be considered.

The analyses in English databases will be carried out in collaboration with PHE, following their guidance, as the RSC is a major data and sampling source for PHE.[65] It will be important to avoid the confusion that might arise from different approaches to analysing RSC data, and to benefit from shared expertise in monitoring VE.[66 67] We have a shared protocol for this season's analysis.[68] Likewise, in Northern Ireland, we will work closely with the Public Health Agency. In Scotland, all planned analyses will also be carried out in collaboration with PHS. In Wales, analyses will be carried out in collaboration with PHW, who are involved with other colleagues in Wales in the national rollout and evaluation of the vaccine deployment, and are unique placed to provide expertise, and are key contributors as part of the existing ConCOV project in Wales.[26]

*Prospective cohort study for clinical outcomes and deaths*

A prospective cohort study will be used to estimate VE against clinical outcomes and deaths in vaccinated and unvaccinated individuals respectively. Time-dependent Cox models will provide the adjusted rate ratios for these outcomes. VE will be calculated according to VE=1 RR. VE estimates will be adjusted for potential confounders and effect modifiers including age, sex, underlying medical condition, SES, history of healthcare utilisation, medication and non-COVID-19 vaccination. Other potential confounders and effect modifiers may also be explored. Recall and misclassification bias will be minimised in our planned prospective cohort studies as we will use data from national linked datasets which allow rapid analysis of vaccination and clinical outcomes data derived from electronic health records. Nevertheless, unmeasured confounding can still influence the VE estimates (given the observational nature of these study designs) despite attempts to provide VE adjusted for potential confounders as mentioned above.

Additional sensitivity or post hoc analyses such as using different time intervals following administration of the vaccine to define exposure will also be explored for all study outcomes related to VE.

#### Vaccine safety
*SCCS and retrospective cohort studies for adverse events*

The risk of any vaccine-related adverse events will be assessed using a SCCS study design.[24] This study design tests whether the risk of an adverse event is higher at post-vaccination period compared with other periods that are temporarily unrelated to vaccine administration.[69] The main advantage of this case series method over other methods of analysis is that it only includes individuals who have been vaccinated. As a result, adequate statistical power can often be obtained with relatively small sample sizes.

In addition, all confounders (eg, sex, genetics, SES, location, underlying condition) that do not vary with time over the observation period are implicitly controlled for.[69] The number of adverse events in a pre-defined risk interval will be compared with predefined control intervals. Risk interval refers to postvaccine administration period over the observation period of the study and control intervals refer to prevaccine and postvaccine administration over the study's observation period. Risk and control intervals will also be determined in relation to vaccine dose administration (eg, between first and second doses of the vaccines). A clearance or wash-out interval between the risk and control intervals will also be applied.

The exact duration (in days) of the risk and control intervals will be determined for each AEFI outcome based on severity level (mild-to-moderate, severe and typical onset) and vaccine type separately (see online supplemental material, appendix 3). We will use the Benjamini-Hochberg procedure to control the False Discovery rate of testing a large number of hypotheses related to each prespecified adverse events of interest.[70] Subgroup analyses by vaccine type, dose and dosing schedules will also be considered.

It is possible that sample selection bias could be induced in the SCCS if inclusion in the study is related nontrivially to the adverse outcome of interest. This may be particularly true for severe adverse events. For example, if an individual has a cardiac arrest then they are less likely to be vaccinated and thus less likely to be included in the study. We will therefore carry out a retrospective cohort or an event-dependent exposure version of SCCS study for severe outcomes.

A sensitivity analysis by previous history of SARS-CoV-2 infection will also be considered. We will explore if previous SARS-CoV-2 infection is associated with any AEs observed following a COVID-19 vaccination.

### Pooled analyses

In our study, we will initially provide estimates on vaccine uptake, effectiveness, and safety for each UK nation. We will also provide pooled estimates across the UK nations. A generic inverse variance method for meta-analysis will be used. Heterogeneity of our pooled estimates will be assessed using the standard $\chi^2$ and the $I^2$ statistic.[71] Forest plots will be used to visualise any statistical heterogeneity in our pooled estimates for the four nations. Individual nation's ORs or RRs and their 95% CIs will be used to estimate the pooled VE estimates. Measured and unmeasured heterogeneity is highly probable across the UK nations. Effect estimates from random-effect models will thus only be considered.

### Sample size

We are basing sample size calculations on Scottish testing and vaccination data because it is currently the only UK nation with full national data coverage at the time of writing. Based on previous work,[72] we estimated a VE of 0.89 against COVID-19 hospitalisation at 28–34 days post-vaccination, with a SD of 0.06. Assuming our VE estimates are asymptotically normally distributed, this gives almost 100% power to detect a VE of over 0.5.

The number of COVID-10 vaccines doses required to detect a relative risk of 5.0 is at least 10 000 doses for a relatively common adverse outcome (eg, myocardial infraction with a background incidence rate of 1400 per 100 000 person years in men older than 85 years old) and more than a million doses are needed to detect a relative risk of 1.5 for a rare adverse outcome (eg, myocardial infraction with a background incidence rate of 28 per 100 000 person years among those 18–34 years).[73 74]

### Patient and public involvement and engagement

Patient and public involvement and engagement (PPIE) members (Antony Chuter, Alex Brownrigg and Jillian Beggs) have been involved since the beginning of this project. The research proposal for the Wales analysis has also been reviewed by members of the public. Their contribution includes defining research questions, interpretation and dissemination of study findings. As part of the ConCOV project[26] in Wales, PPIE members have been involved in the design and evaluation of findings, as well as presentations made to the independent SAIL consumer panel group made up of lay members.

### ETHICS AND DISSEMINATION

For Scotland, approvals have been obtained by the National Research Ethics Service Committee (REC), South East Scotland 02 (REC number: 12/SS/0201) and the Public Benefit and Privacy Panel for Health and Social Care (reference number: 1920–0279). For Wales, the data used in this study are available in the SAIL Databank at Swansea University, Swansea, UK. All proposals to use SAIL data are subject to review by an independent Information Governance Review Panel (IGRP). Before any data can be accessed, approval must be given by the IGRP. The IGRP gives careful consideration to each project to ensure proper and appropriate use of SAIL data. When access has been approved, it is gained through a privacy-protecting safe haven and remote access system referred to as the SAIL Gateway. SAIL has established an application process to be followed by anyone who would like to access data via SAIL.[27] Similarly, in Northern Ireland, the project was approved by the HSC HBS Governance Board (project number 0064), and accessed through the online secure research platform. Findings will be presented at conferences, published in peer-reviewed journals and to the funders and government COVID-19 advisory bodies as appropriate. Strengthening the Reporting of Observational Studies in Epidemiology and Reporting of studies Conducted using Observational Routinely-collected Data (via the COVID-19 extension) checklists will guide our study findings reporting.[75 76] We will also consider using the European Network of Centres for Pharmacoepidemiology and Pharmacovigilance checklist.[77]

**Author affiliations**
[1] The University of Edinburgh, Usher Institute, Edinburgh, UK
[2] Department of Mathematics and Statistics, University of Strathclyde, Glasgow, UK
[3] Public Health Scotland, Glasgow, UK
[4] Nuffield Department of Primary Care Health Sciences, University of Oxford, Oxford, UK
[5] School of Medicine, Dentistry and Biomedical Sciences, Queen's University Belfast, Belfast, UK
[6] BREATHE – The Health Data Research Hub for Respiratory Health, London, UK
[7] Population Data Science, Swansea University Medical School, Swansea, UK
[8] Vaccine Preventable Disease Programme, Public Health Wales, Cardiff, UK
[9] Health Data Research, London, UK
[10] School of Medicine, University of St Andrews, St Andrews, UK
[11] Public Health Scotland, Edinburgh, UK
[12] MRC/CSO Social and Public Health Sciences Unit, University of Glasgow, Glasgow, UK
[13] Wellington School of Health, Faculty of Health, Victoria University of Wellington, Wellington, New Zealand

**Acknowledgements** England: Patients and general practices who provide samples and agree to share data with the Oxford-RCGP RSC. EMIS, TPP, In-Practice Systems, Wellbeing software for supporting the RSC data extraction. Public Health England for support and collaboration. Wales: This work uses data provided by patients and collected by the NHS as part of their care and support. We would also like to acknowledge all data providers who make anonymised data available for research. We wish to acknowledge the collaborative partnership that enabled acquisition and access to the de-identified data, which led to this output. The collaboration was led by the Swansea University Health Data Research UK team under the direction of the Welsh Government Technical Advisory Cell (TAC) and includes the following groups and organisations: the Secure Anonymised Information Linkage (SAIL) Databank, Administrative Data Research (ADR) Wales, NHS Wales Informatics Service (NWIS), Public Health Wales, NHS Shared Services Partnership and the Welsh Ambulance Service Trust (WAST). All research conducted has been completed under the permission and approval of the SAIL independent Information Governance Review Panel (IGRP) project number 0911. Northern Ireland: The authors would like to acknowledge the help provided by the staff of the Honest Broker Service (HBS) within the Business Services Organisation Northern Ireland (BSO). The HBS is funded by the BSO and the Department of Health (DoH). The authors alone are responsible for the interpretation of the data and any views or opinions presented are solely those of the author and do not necessarily represent those of the BSO.

**Contributors** AS conceived this study and commented on several drafts of the protocol. EV wrote the first draft of the protocol with assistance from TS and SK. SdeL, RT, MJ and DMcC, JW and RH contributed about the English sentinel system and commented on the research methods. DB, DO and SM contributed about the Northern Ireland data and commented on the research methods. AC and JB were involved in the study design and commented on introduction and research methods from a patient and public perspective. DF, CO, AA, SB, GD, LJG, RG, EL, JL, RAL, LN, MP and FT contributed about the Wales data and commented on the research methods. JP contributed on the study design and commented on several drafts. JM, CM, UA, RW, SJS, EM, PH, CRS and CR contributed about the Scotland data and comments on research methods. All authors contributed to the study design. All authors contributed to drafting the protocol. All authors revised the manuscript for important intellectual content. All authors gave final approval of the version to be published.

**Funding** This research is part of the Data and Connectivity National Core Study, led by Health Data Research UK in partnership with the Office for National Statistics and funded by UK Research and Innovation (HDRUK2020.146). EAVE II is funded by the Medical Research Council (MC_PC_19075) and supported by the Scottish Government. This work is supported by BREATHE - The Health Data Research Hub for Respiratory Health (MC_PC_19004). BREATHE is funded through the UK Research and Innovation Industrial Strategy Challenge Fund and delivered through Health Data Research UK. ConCOV is supported by the Medical Research Council (MR/V028367/1); Health Data Research UK (HDR-9006) which receives its funding from the UK Medical Research Council, Engineering and Physical Sciences Research Council, Economic and Social Research Council, Department of Health and Social Care (England), Chief Scientist Office of the Scottish Government Health and Social Care Directorates, Health and Social Care Research and Development

Division (Welsh Government), Public Health Agency (Northern Ireland), British Heart Foundation (BHF) and the Wellcome Trust; and Administrative Data Research UK which is funded by the Economic and Social Research Council (grant ES/S007393/1).

**Competing interests** AS is a member of the Scottish Government Chief Medical Officer's COVID-19 Advisory Group. RAL reports grants from MRC during the conduct of the study. SJS reports grants from Wellcome Trust, during the conduct of the study; grants from National Institute of Healthcare Research HTA, grants from Tommy's Charity and grants from Chief Scientist for Scotland, outside the submitted work. SdeL is Director of the Royal College of General Practitioners Research and Surveillance Centre. He has received grant funding through his University from AstraZeneca, Eli Lilly, GSK MSD, Seqirus and Takeda. He has been members of advisory boards for AstraZeneca, Sanofi, and Seqirus. DB is jointly employed by Queen's University Belfast, the Public Health Agency and the Department of Health (Northern Ireland), and he is currently or has been a member of COVID-19 government advisory groups, including the Scientific Advisory Group for Emergencies (SAGE), its subgroups, and the UK Vaccine Effectiveness Expert Panel. All other authors report no conflicts of interest.

**Patient consent for publication** Not applicable.

**Provenance and peer review** Not commissioned; externally peer reviewed.

**ORCID iDs**
Eleftheria Vasileiou http://orcid.org/0000-0001-6850-7578
Ting Shi http://orcid.org/0000-0002-4101-4535
Ruby Tsang http://orcid.org/0000-0002-2520-526X
Simon de Lusignan http://orcid.org/0000-0001-5613-6810
Chris Orton http://orcid.org/0000-0002-9561-2493
Ashley Akbari http://orcid.org/0000-0003-0814-0801
Lucy J Griffiths http://orcid.org/0000-0001-9230-624X
Emily Lowthian http://orcid.org/0000-0001-9362-0046
Jane Lyons http://orcid.org/0000-0002-4407-770X
Ronan A Lyons http://orcid.org/0000-0001-5225-000X
Rachael Wood http://orcid.org/0000-0003-4453-623X
Sarah Jane Stock http://orcid.org/0000-0003-4308-856X
Paul Henery http://orcid.org/0000-0003-0380-738X
Aziz Sheikh http://orcid.org/0000-0001-7022-3056

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
