## [Reviewer comments · BMJ Open]

ARTICLE DETAILS

TITLE (PROVISIONAL)	Investigating the uptake, effectiveness and safety of COVID-19 vaccines: protocol for an observational study using linked UK national data
AUTHORS	Vasileiou, Eleftheria; Shi, Ting; Kerr, Steven; Robertson, Chris; Joy, Mark; Tsang, Ruby; McGagh, Dylan; Williams, John; Hobbs, Richard; de Lusignan, Simon; Bradley, Declan; O'Reilly, Dermot; Murphy, Siobhan; Chuter, Antony; Beggs, Jillian; Ford, David; Orton, Chris; Akbari, Ashley; Bedston, Stuart; Davies, Gareth; Griffiths, Lucy; Griffiths, Rowena; Lowthian, Emily; Lyons, Jane; Lyons, Ronan; North, Laura; Perry, Malorie; Torabi, Fatemeh; Pickett, James; McMenamin, Jim; McCowan, Colin; Agrawal, Utkarsh; Wood, Rachael; Stock, Sarah; Moore, Emily; Henery, Paul; Simpson, Colin; Sheikh, Aziz

VERSION 1 – REVIEW

REVIEWER	Debabrata Roy Drug Safety Research Unit
REVIEW RETURNED	28-May-2021

GENERAL COMMENTS	This observational study will provide evidence on the utilisation, effectiveness and safety of the COVID-19 vaccines in the UK and is of very high interest. Please find my comments on the protocol: - Overall the manuscript is very comprehensive in parts but requires restructuring, consistency and further details in parts to improve readability.- It would be helpful to the reader for the utilisation, effectiveness and safety elements of the study to be presented separately, where relevant, in every section e.g. Introduction, Method etc.- In addition, further distinction should be clearly made in the manuscript, where relevant, for each nation in the UK- Methodologies used for different analyses in each element of the study should be presented separately within the relevant section- Separate sections where relevant (safety study related sections) may also be required for each COVID-19 vaccine being examined.- There are sections of the protocol which would benefit from being more concise and other sections where further information is required, as well as keywords which may be missing or additional words needing removal. Please could this be addressed in the review?- The introduction does not present information on adverse events related to COVID-19 vaccinations from clinical trials as well as from spontaneous reports in the post-marketing setting e.g. thrombosis with thrombocytopenia or myocarditis. This study has the potential to quantify potential signals which are detected via the spontaneous reporting system so relevant signals should be reviewed and included as adverse events of interest in the study.
---

	 - The introduction includes utilisation information from a US survey. Would results from a UK survey be more appropriate? - The aims and objectives could be presented in a separate section, with primary and secondary objectives of the study. - Would GP surgeries using VISION software be using Read codes, and would these codes be used in England? Coding conventions have not been stated for Northern Ireland. - Could a statement be added on how the algorithms for case definition will be developed. Also the process for case validation. - The exposure definition would require re-wording as it is unclear how this is defined. it is unclear why time periods are labelled as unexposed or exposed, and whether this should be based on the date of vaccine administration to the patient? - Sliding window will need further detail please? - I am unsure what is meant by individual-level analyses and whether this should be reworded? - For the SCCS method if a patient dies then by definition they will not be included in the study, so may not be the most appropriate analogy. To the best of my knowledge severe AE's can be outcomes in the SCCS design but these must be acute events and we must have knowledge of the risk interval following vaccination. - I am unsure if the FDR method (Benjamini-Hochberg) is required as adverse events of interest can be examined independently? - Sample size/power calculations would also be required for the safety study and will be dependent on the background rates of the AE's of interest. - Will any sensitivity analyses be performed for the effectiveness and safety study aspects? - Strengths and limitations sections for each study element would be useful. - Are you considering the use of a buffer/wash-out period between the risk interval and the control interval? - The Moderna COVID-19 vaccine has been authorised for use in the UK so this information could be updated in the introduction. Some information on UK COVID-19 vaccine delivery strategies since December 2020, and recent age based guidance on vaccine choice based on the benefit-risk assessments and spontaneous reports would be useful. - For prior and concomitant medications, I would also be interested in NOACs, warfarin and heparin use if feasible. - Would ENCEPP guidelines be worth referring to? - Would effectiveness be more appropriate than protection levels? - Would further ethical approval information be required for genome sequencing data? - Further information on the unvaccinated comparator cohort in the cohort study for each COVID-19 vaccine would be required e.g. censoring criteria.
--	--

REVIEWER	Robert P Lennon Penn State College of Medicine, Family and Community Medicine
REVIEW RETURNED	05-Jul-2021

GENERAL COMMENTS	Bmjopen-2021-050062 Authors, very well done. Please consider some minor changes for clarity, detailed below. Abstract methods line 35-36. "VE estimates on against laboratory confirmed. . ." The "on against" language is not clear to me. Do you
---

	mean, “VE estimates will be generated from laboratory confirmed SARS-CoV-2 infection using a generalised additive logistic model”? Or just cut “on”? P5, line 14-15. Please update the date from 07 February 2021 – likely to pass 4 million deaths by the end of this month. No criticism – just changing fast enough to warrant an update before publication. P5, line 20: “. . . their pre-licensure clinical trials and approved for. . .” Add, “been” between ‘and’ and ‘approved’. P5, line 36: as above, consider updates the dose and population number. P7 lines 6-7: “. . . collected at primary, secondary, linked with mortality. . .” Do you mean, “. . . collected at primary and secondary care visits, linked with mortality. . .”? Please rephrase for clarity. P7, lines 10-11: “Pooled estimates across the UK national will also be calculated.” Across national data? Across nations? Across national samples? P7, lines 14-15: Why do you estimate 30% coverage? Please give a citation in support or explain your reasoning. P7, line 35: “. . . study ID in analysis dataset.” □ “. . . study ID in the analysis dataset.” P8, line 14-16: “. . . variables are those for COVID-19, there have . . .” Semi-colon instead of comma after COVID-19. P8, Lines 56-57: “Data is are routinely extracted. . .” Please choose either ‘is’ or ‘are’. While data is a plural noun, hence for academic settings “are” is indicated, when used like this as a singular mass noun is now widely accepted to use the singular. Defer to editors for choice. Methods, general: In the UK is there much private testing? In the US this is a growing phenomenon, and may prove difficult to track. If there is the potential for significant testing without the ability to capture results, please mention how this will be addressed. If all (or virtually all) testing will be captured in these systems, there is no need to address this comment. P14, lines 35—36: you mention confounders that do not vary with time over the observation period; will your data be granular enough to determine if SES, location, and underlying conditions change over the observation period? How will those knowns or unknowns impact your results or interpretation? P15, lines 34-35: “Research proposal for the Wales . . .” Either ‘A research proposal’ or ‘The research proposal’.
--	--

VERSION 1 – AUTHOR RESPONSE

Reviewer #1

This observational study will provide evidence on the utilisation, effectiveness and safety of the COVID-19 vaccines in the UK and is of very high interest.

Response: Thank you.

1. *Overall the manuscript is very comprehensive in parts but requires restructuring, consistency and further details in parts to improve readability.*

Response: We have now made all suggested revisions which we believe have improved the readability of our protocol.

2. *It would be helpful to the reader for the utilisation, effectiveness and safety elements of the study to be presented separately, where relevant, in every section e.g. Introduction, Method etc.*

Response: We have now presented (where relevant) these sections related to the analyses on vaccine uptake, effectiveness and safety separately throughout of the protocol.

3. *In addition, further distinction should be clearly made in the manuscript, where relevant, for each nation in the UK.*

Response: We have now created (where relevant) distinct sections in our protocol for each UK nation.

4. *Methodologies used for different analyses in each element of the study should be presented separately within the relevant section*

Response: Study designs used for different study outcomes are now presented separately (see pages 15-17).

5. *Separate sections where relevant (safety study related sections) may also be required for each COVID-19 vaccine being examined.*

Response: Given the ongoing evidence related to various adverse events following certain vaccines, we feel that assessing the safety of all vaccines against all potential adverse events would be invaluable to policy makers. Our preference is therefore to assess all vaccine types and doses against any potential reported adverse effects. We nonetheless recognise that there may be instances in which it is more appropriate to focus on a particular vaccine and have now made this point in the revised protocol (see page 13):

“AEFI by specific vaccine type will also be considered.”

6. *There are sections of the protocol which would benefit from being more concise and other sections where further information is required, as well as keywords which may be missing or additional words needing removal. Please could this be addressed in the review?*

Response: We have addressed these issues in our revised version. For example, the introduction is now updated and contains more relevant information (see pages 5-7). In addition, the section around exposure definitions of our methods is now more concise (see page 12). Relevant keywords have also been added and redundant words have been removed.

7. *The introduction does not present information on adverse events related to COVID-19 vaccinations from clinical trials as well as from spontaneous reports in the post-marketing setting e.g. thrombosis with thrombocytopenia or myocarditis. This study has the potential to quantify potential signals which are detected via the spontaneous reporting system so relevant signals should be reviewed and included as adverse events of interest in the study.*

Response: Additional information related to adverse events following COVID-19 vaccinations has now been added to our revised Introduction and reads as follows (see page 5):

“All three vaccines were well tolerated with mild-to-moderate side-effects mostly reported.[6,7,8] In March 2021, the European Medicines Agency (EMA) reported extremely rare but serious side-effects including blood clots and bleeding following administration of the Oxford-AstraZeneca vaccine.[9,10] As a result, the Joint Committee on Vaccination and Immunisation (JCVI) of the UK Government has recommended that healthy adults aged 18-39 years olds should not be offered the Oxford-AstraZeneca vaccine but the Pfizer-BioNTech or Moderna vaccines instead.[11] Rare cases of cardiac inflammation including myocarditis and pericarditis following immunisation with the Pfizer-BioNTech and Moderna vaccines have also been reported [12] and are thus closely monitored by national and international regulatory agencies.”

8. *The introduction includes utilisation information from a US survey. Would results from a UK survey be more appropriate?*

Response: Results from the US survey are now removed, and the introduction now includes the following results from a UK survey instead (see page 6):

“In the UK, the ‘Understanding Society’ COVID-19 survey asked 12,035 participants (in November 2020) their likelihood of vaccine uptake and reason for hesitancy.[19] High levels of hesitancy were found in women (21%), younger adults aged 16-24 years olds (27%), those with lower education levels (19%) and in Black (72%) and Pakistani/Bangladeshi (42%) ethnic groups.[19] This is particularly concerning given that ethnic minority groups are some of the subgroups of the population that have been most at risk in this pandemic.[19]”

9. *The aims and objectives could be presented in a separate section, with primary and secondary objectives of the study.*

Response: The aim and objectives of the study are now provided in a separate section and read as follows (see page 6-7):

“Aim and objectives

The aims of this study are to assess the uptake, effectiveness, and safety of the currently licenced COVID-19 vaccines (Pfizer-BioNTech, Oxford-AstraZeneca and Moderna) in the UK using linked healthcare and administrative data. We will also seek to assess any additional licensed vaccines during the course of this study.

Our primary objectives are to: a) measure variation in vaccine uptake in relation to population characteristics; b) assess vaccine effectiveness (VE) against infection, transmission, [20] severe outcomes, and deaths; and c) identify the risk of adverse events following immunisation (AEFIs) in each UK nation. Our secondary objectives are to provide UK-wide pooled estimates of each primary objective.”

10. *Would GP surgeries using VISION software be using Read codes, and would these codes be used in England? Coding conventions have not been stated for Northern Ireland.*

Response: In Scotland and Wales, Vision and EMIS are the current providers of IT in GP surgeries and Read codes is the recommended national standard coding system in GPs for recording clinical information.[1,2] In England, data from GPs are recorded using Systematised Nomenclature of Medicine (SNOMED) Clinical Terms (CT) codes.[3]

In Northern Ireland, Read codes are used to record any clinical information from GP practices. However, for this project, we are still in the process of getting access to GP data. We are thus unable to mention additional details regarding to GP data at this stage.

[1] <https://www.isdscotland.org/Health-Topics/General-Practice/GP-Consultations/Grouping-clinical-codes.asp>

[2] <https://www.isdscotland.org/health-topics/general-practice/spire/>

[3] <https://www.england.nhs.uk/digitaltechnology/digital-primary-care/snomed-ct/>

11. *Could a statement be added on how the algorithms for case definition will be developed. Also the process for case validation.*

Response: Case definitions are now included and read as follows (see page 13):

“A COVID-19 hospital or ICU admission will be defined based on either a RT-PCR confirmed positive test for SARS-CoV-2 in the 28 days prior to admission or based on an ICD-10 code for COVID-19 (U07.1 or U07.2) in any diagnostic position. A COVID-19 death will be defined as COVID-19 as the underlying ICD-10 cause of death recorded on the death certificate, or death from any cause within 28 days of a positive RT-PCR test for SARS-CoV-2.”

To the best of our knowledge, there is not a standard process for case validation. However, we accept that positive tests and coded hospitalisations and deaths are accurate.

12. *The exposure definition would require re-wording as it is unclear how this is defined. it is unclear why time periods are labelled as unexposed or exposed, and whether this should be based on the date of vaccine administration to the patient?*

Response: Our exposure definition is now updated and reads as follows (see page 12):

“For the first vaccine dose (partial vaccination), an individual will be defined as exposed or vaccinated from day 14 after receiving the first dose between the period of 8 December 2020 and until the end of follow up. For the second vaccine dose (full vaccination), an individual will be defined as exposed or vaccinated from day 14 after receiving the second dose during the study period.

Exposed or vaccinated groups will be stratified by the following time intervals:

a) 0-13 days after dose 1; b) 14-20 days after dose 1; c) 21-27 days after dose 1; d) 28-34 days after dose 1; e) 35-41 days after dose 1; f) ≥ 42 days after dose 1; g) 0-13 days after dose 2 and h) ≥ 14 days after dose 2.”

13. *Sliding window will need further detail please?*

Response: The following further detail is now provided (see page 11):

“In England, we will use a customised “sliding window” to capture pregnancy data.[49] Specifically, an algorithm that accurately inferred pregnancies will be used by adopting an ontological approach for case finding.[49] The ontological approach will thus be used to identify pregnancies and associated

complications using a systematic approach to derive this information from routinely collected administrative health data which will be available via the RCGP RSC.[49]”

14. I am unsure what is meant by individual-level analyses and whether this should be reworded?

Response: Individual-level analyses refer to analyses using data on individuals. Specifically, individual patient-level data refers to any information (e.g., study exposure, outcome, or other individual characteristics) related to every single patient in our study datasets. Therefore, our analyses will be at individual patient-level since all study information will derive from patients rather than combining data from each patient and providing an average number of a characteristic (e.g., age etc) which is the case of aggregate data. We have thus amended this term to ‘Individual patient-level analyses’ in page 15.

15. For the SCCS method if a patient dies then by definition they will not be included in the study, so may not be the most appropriate analogy. To the best of my knowledge severe AE's can be outcomes in the SCCS design but these must be acute events and we must have knowledge of the risk interval following vaccination.

Response: We agree with the reviewer’s comment regarding the death outcome, and we have thus amended our text as follows (see page 17):

“For example, if an individual has a cardiac arrest, then they are less likely to be vaccinated and thus less likely to be included in the study.”

16. I am unsure if the FDR method (Benjamini-Hochberg) is required as adverse events of interest can be examined independently?

Response: We believe that the Benjamini-Hochberg False Discovery rate will be useful given the large number of adverse events that we will consider as outcomes of interest. Adjustment for multiple testing via the Benjamini-Hochberg may thus be required.

17. Sample size/power calculations would also be required for the safety study and will be dependent on the background rates of the AE's of interest.

Response: Sample size calculations are now updated for VE estimates and also provided for adverse events and read as follows (see page 17):

“We are basing sample size calculations on Scottish testing and vaccination data because it is currently the only UK nation with full national data coverage. Based on previous work,[72] we estimated a VE of 0.89 against COVID-19 hospitalisation at 28-34 days post vaccination, with a standard deviation of 0.06. Assuming our VE estimates are asymptotically normally distributed, this gives almost 100% power to detect a VE of over 0.5.

The number of COVID-10 vaccines doses required to detect a relative risk of 5.0 is at least 10,000 doses for a relatively common adverse outcome (e.g., myocardial infraction with a background incidence rate of 1400 per 100,000 person years in men older than 85 years old) and more than a million doses are needed to detect a relative risk of 1.5 for a rare adverse outcome (e.g., myocardial infraction with a background incidence rate of 28 per 100,000 person years among those 18-34 years).[73, 74]”

18. Will any sensitivity analyses be performed for the effectiveness and safety study aspects?

Response: Sensitivity analyses for the vaccine effectiveness and safety studies are now provided in the following paragraphs (see pages 16 & 17):

“Additional sensitivity or post-hoc analyses such as using different time intervals following administration of the vaccine to define exposure will also be explored for all study outcomes related to vaccine effectiveness.”

“A sensitivity analysis by previous history of SARS-CoV-2 infection will also be considered. We will explore if previous SARS-CoV-2 infection is associated with any AEs observed following a COVID-19 vaccination.”

19. Strengths and limitations sections for each study element would be useful.

Response: the following strengths and limitations are now provided (see page 15-17):

“The main advantage of the TND studies compared to traditional case-control studies is that it minimises confounding factors from health care-seeking behaviour, which means both cases and controls have similar likelihood of seeking healthcare when having symptoms indicative of COVID-19 illness.[64] Selection bias can still arise if study participants are not recruited based on pre-defined criteria (e.g., signs/symptoms indicative to COVID-19 illness) but based on clinician-ordered test.[64] In this scenario, clinicians may be more likely to carry out a test on patients that are more likely to have COVID-19 illness (outcome) or not being vaccinated.[64] This will result to biased sampling (non-representativeness) of the study participants from the source population which could lead to overestimation of the VE estimates.[64]”

“Recall and misclassification bias will be minimised in our planned prospective cohort studies as we will use data from national linked datasets which allow rapid analysis of vaccination and clinical outcomes data derived from electronic health records. Nevertheless, unmeasured confounding can still influence the VE estimates (given the observational nature of these study designs) despite attempts to provide VE adjusted for potential confounders as mentioned above.

“The main advantage of this case series method over other methods of analysis is that it only includes individuals who have been vaccinated. As a result, adequate statistical power can often be obtained with relatively small sample sizes. In addition, all confounders (e.g., sex, genetics, SES, location, underlying condition) that do not vary with time over the observation period are implicitly controlled for.[69]”

“It is possible that sample selection bias could be induced in the SCCS if inclusion in the study is related nontrivially to the adverse outcome of interest. This may be particularly true for severe adverse events.”

20. Are you considering the use of a buffer/wash-out period between the risk interval and the control interval?

Response: We will consider a wash-out or clearance period between the risk interval and the control interval and have thus provided the following text (see page 16):

“A clearance or wash-out interval between the risk and control intervals will also be applied.”

21. The Moderna COVID-19 vaccine has been authorised for use in the UK so this information could be updated in the introduction. Some information on UK COVID-19 vaccine delivery strategies since December 2020, and recent age based guidance on vaccine choice based on the benefit-risk assessments and spontaneous reports would be useful.

Response: We have now updated our introduction related to vaccines and provided the following text (see page 5):

“As of the 8 January 2021, the UK has also approved a third COVID-19 vaccine manufactured by Moderna which has shown 94% efficacy against confirmed SARS-CoV-2 infection after receipt of second dose.[8] All three vaccines were well tolerated with mild-to-moderate side-effects mostly reported.[6,7,8] In March 2021, the European Medicines Agency (EMA) reported extremely rare but serious side-effects including blood clots and bleeding following administration of the Oxford-AstraZeneca vaccine.[9,10] As a result, the Joint Committee on Vaccination and Immunisation (JCVI) of the UK Government has recommended that healthy adults aged 18-39 years olds should not be offered the Oxford-AstraZeneca vaccine but the Pfizer-BioNTech or Moderna vaccines instead.[11] Rare cases of cardiac inflammation including myocarditis and pericarditis following immunisation with the Pfizer-BioNTech and Moderna vaccines have also been reported [12] and are thus closely monitored by national and international regulatory agencies.

“As of 20 September 2021, over 48 million people have received their first vaccine dose and over 44 million people have received their second vaccine dose across the UK [13] based on JCVI’s vaccination priority list which targets those most at risk of severe COVID-19 illness (e.g., older adults and people with comorbidities).[14]”

22. For prior and concomitant medications, I would also be interested in NOACs, warfarin and heparin use if feasible.

Response: We have now included these medications and provided the following (see page 14):

“Prior or concomitant usage of the novel oral anticoagulants (NOACs), warfarin and heparin will also be measured.”

23. Would ENCEPP guidelines be worth referring to?

Response: We will consider using the ENCEPP guidelines and we have thus mentioned the following text (see page 18):

“We will also consider using the European Network of Centres for Pharmacoepidemiology and Pharmacovigilance checklist.[77]”

However, for this project, we will be using the Strengthening the Reporting of Observational Studies in Epidemiology (STROBE) and Reporting of studies Conducted using Observational Routinely-collected Data (RECORD) (via the COVID-19 extension) checklists for our study protocol and when reporting study results given that the ENCePP checklist is not listed in the Enhancing the QUALity and Transparency Of health Research (EQUATOR) network.

24. *Would effectiveness be more appropriate than protection levels?*

Response: The sentence is now updated and reads as follows (see page 6):

“Specifically, robust observational epidemiological studies are required to measure coverage rates in the population in relation to demographic and other population characteristics, assess effectiveness against infection, severe illness and deaths, and to detect adverse events.”

25. *Would further ethical approval information be required for genome sequencing data?*

Response: Our current ethical approvals already cover access to genome sequencing data. No additional approvals are thus required.

26. *Further information on the unvaccinated comparator cohort in the cohort study for each COVID-19 vaccine would be required e.g. censoring criteria.*

Response: Additional information of the unvaccinated/controls is now provided and reads as follows (see page 12):

“Controls or unvaccinated will be defined as those who have not yet received a COVID-19 vaccine or have only received one vaccine dose. Controls who become vaccinated with any vaccine (i.e., including one dose of Moderna) or receive a second vaccine will then be assigned within the exposure group. As a result, follow-up of the exposure period will be censored for both the vaccinated and control recipient if the control meets the criteria to be classified as exposed (receiving a first dose when compared with the unvaccinated group and receiving a second dose when compared with the partially/one dose vaccinated group). Maximum follow-up period will correspond to the latest event date depending on the outcome of interest.”

Reviewer #2

Authors, very well done. Please consider some minor changes for clarity, detailed below.

Response: Thank you.

1. *Abstract methods line 35-36. “VE estimates on against laboratory confirmed. . .” The “on against” language is not clear to me. Do you mean, “VE estimates will be generated from laboratory confirmed SARS-CoV-2 infection using a generalised additive logistic model”? Or just cut “on”?*

Response: We have now removed the word “on” and sentence reads as follows (see page 3):

“VE estimates against laboratory confirmed SARS-CoV-2 infection will be generated using a generalised additive logistic model.”

2. *P5, line 14-15. Please update the date from 07 February 2021 – likely to pass 4 million deaths by the end of this month. No criticism – just changing fast enough to warrant an update before publication.*

Response: We have now updated the following sentence (see page 5):

“On 11 March 2020, the World Health Organization (WHO) declared the coronavirus 2019 disease (COVID-19) as a global pandemic, which as of 20 September 2021 has caused more than 228 million infections and four million deaths worldwide.[1]”

3. P5, line 20: “. . . their pre-licensure clinical trials and approved for. . .” Add, “been” between ‘and’ and ‘approved’.

Response: The sentence is now updated and reads as follows (see page 5):

“Dozens of COVID-19 vaccines are currently under development with some vaccines now having successfully completed their pre-licensure clinical trials and been approved for population vaccine administration.[2,3]”

4. P5, line 36: as above, consider updates the dose and population number.

Response: The vaccine dose and population number in the UK is now updated as shown below (see page 5):

“As of 20 September 2021, over 48 million people have received their first vaccine dose and over 44 million people have received their second vaccine dose across the UK [13] based on JCVI’s vaccination priority list which targets those most at risk of severe COVID-19 illness (e.g., older adults and people with comorbidities).[14]”

5. P7 lines 6-7: “. . . collected at primary, secondary, linked with mortality. . .” Do you mean, “. . . collected at primary and secondary care visits, linked with mortality. . .”? Please rephrase for clarity.

Response: The sentence is now rephrased as seen below (see page 7):

“We will use pseudonymised individual level data routinely collected at primary and secondary care visits, linked with mortality, laboratory, and vaccination data across the UK.”

6. P7, lines 10-11: “Pooled estimates across the UK national will also be calculated.” Across national data? Across nations? Across national samples?

Response: The sentence is now corrected and reads as follows (see page 7):

“Pooled estimates across the UK nations will also be calculated.”

7. P7, lines 14-15: Why do you estimate 30% coverage? Please give a citation in support or explain your reasoning.

Response: This was a typo in the original manuscript the estimated 30% coverage related to primary care data not national coverage. We have amended the manuscript to state that national coverage in Northern Ireland will be approximately 1.9 million people (see page 7).

8. P7, line 35: “. . . study ID in analysis dataset.” □ “. . . study ID in the analysis dataset.”

Response: Sentence is now updated and reads as follows (see page 8):

“The Health Care Number (HCN) will be used to link individuals’ data though replaced with an anonymous study ID in the analysis dataset.”

9. *P8, line 14-16: “. . . variables are those for COVID-19, there have . . .” Semi-colon instead of comma after COVID-19.*

Response: Sentence is now updated as shown below (see page 8):

“A key part of these curated variables are those for COVID-19; there have been three iterations of these and we have carefully curated case definitions.[34-36]”

10. *P8, Lines 56-57: “Data is are routinely extracted. . .” Please choose either ‘is’ or ‘are’. While data is a plural noun, hence for academic settings “are” is indicated, when used like this as a singular mass noun is now widely accepted to use the singular. Defer to editors for choice.*

Response: Sentence is now removed as it is no longer applicable (see page 9).

11. *Methods, general: In the UK is there much private testing? In the US this is a growing phenomenon, and may prove difficult to track. If there is the potential for significant testing without the ability to capture results, please mention how this will be addressed. If all (or virtually all) testing will be captured in these systems, there is no need to address this comment.*

Response: In the UK, the majority of laboratory tests for COVID-19 are conducted in public sector primary or secondary care settings. Therefore, the results from these tests will be captured in our study datasets.[1]

[1] <https://www.gov.uk/government/publications/coronavirus-covid-19-testing-data-methodology/covid-19-testing-data-methodology-note>

12. *P14, lines 35—36: you mention confounders that do not vary with time over the observation period; will your data be granular enough to determine if SES, location, and underlying conditions change over the observation period? How will those knowns or unknowns impact your results or interpretation?*

Response: Yes, we will be able to have individual patient-level data and thus we will be able to assess if a patient’s SES, location or underlying condition changes over time. The observation period of our planned SCCS study design will be relatively short with pre-defined risk and control intervals expanding a few months. We therefore do not expect a significant change of the SES, location, underlying conditions or other patient characteristics that will have a potential significant impact when interpreting our study results.

13. *P15, lines 34-35: “Research proposal for the Wales . . .” Either ‘A research proposal’ or ‘The research proposal’.*

Response: The sentence is now corrected and reads as follows (see page 18):

“The research proposal for the Wales analysis has also been reviewed by members of the public.”